# Dietary Supplemented Curcumin Improves Meat Quality and Antioxidant Status of Intrauterine Growth Retardation Growing Pigs via Nrf2 Signal Pathway

**DOI:** 10.3390/ani10030539

**Published:** 2020-03-24

**Authors:** Ligen Zhang, Jiaqi Zhang, Enfa Yan, Jintian He, Xiang Zhong, Lili Zhang, Chao Wang, Tian Wang

**Affiliations:** College of Animal Science and Technology, National Experimental Teaching Demonstration Centre of Animal Science, Nanjing Agricultural University, Nanjing 210095, China; zltaoqi@gmail.com (L.Z.); 2019105063@njau.edu.cn (J.Z.); 2018805112@njau.edu.cn (E.Y.); 15150537273@163.com (J.H.); zhongxiang@njau.edu.cn (X.Z.); zhanglili@njau.edu.cn (L.Z.);

**Keywords:** curcumin, intrauterine growth retardation, meat quality, antioxidant capacity, Nrf2, mineral

## Abstract

**Simple Summary:**

More than 15% of piglets and about 10% of newborn humans suffer from intrauterine growth retardation (IUGR), which refers to growth lag, developmental restriction and impaired organs in the fetus. IUGR exhibits programming consequences and exerts permanent negative effects on postnatal growth and health. Dietary supplemented curcumin, as the main natural polyphenol isolated from the natural antioxidant (turmeric), might show possible effects on antioxidant capacity, and the meat quality of IUGR pigs. Therefore, in our present study, 12 normal birth weight (NBW) and 24 IUGR neonatal female piglets were selected and fed control diets supplemented 0 (NBW), 0 (IUGR) and 200 (IUGR + Cur) mg/kg curcumin from 26 to 115 days of age (n = 12). The growth performance, meat quality, redox status and its related Nrf2 pathway were determined to test the hypothesis that curcumin may play beneficial roles against IUGR-induced oxidative stress. This study suggested that curcumin could serve as a potential natural antioxidant in nutrition interventions of IUGR offspring to enhance the redox status and improve the meat quality of leg muscles. These results attained from IUGR pig models can also provide some useful theoretical references for IUGR offspring in humans.

**Abstract:**

Intrauterine growth retardation (IUGR) exhibits programming consequences and may induce oxidative stress in growing animals and humans. This study was conducted to investigate the hypothesis that dietary curcumin may protect growing pigs from IUGR-induced oxidative stress via the Nrf2 pathway. Twelve normal birth weight (NBW) and 24 IUGR female piglets were selected and fed control diets supplemented 0 (NBW), 0 (IUGR) and 200 (IUGR + Cur) mg/kg curcumin from 26 to 115 days of age (n = 12). Growth performance, meat quality, redox status and its related Nrf2 pathway were determined. Results showed that IUGR pigs exhibited decreased body weight on 0 d, 26 d and 56 d (*p* < 0.01) but had no difference on 115 d among NBW, IUGR and IUGR + Cur groups (*p* > 0.05). Compared with NBW and IUGR groups, a significant decrease in drip loss (24 h and 48 h) was observed in the IUGR + Cur group (*p* < 0.01). IUGR pigs had higher concentrations of malondialdehyde (MDA) (*p* < 0.01) and protein carbonyl (PC) (*p* = 0.03) and lower activities of glutathione peroxidase (*p* = 0.02), catalase (*p* < 0.01) and peroxidase (*p* = 0.02) in leg muscles than NBW pigs. Dietary-added 200 mg/kg curcumin decreased concentrations of MDA and PC and improved the activities of catalase, superoxide dismutase (SOD) and peroxidase as compared to the IUGR group (*p* < 0.05). Additionally, dietary curcumin enhanced protein (NQO1) and mRNA expression of genes (Nrf2, NQO1, gamma-glutamyltransferase 1 (GGT1), heme oxygenase-1 (HO-1), glutathione S-transferase (GST) and catalase (CAT)) as compared to the IUGR group (*p* < 0.05). These results suggest that dietary curcumin could serve as a potential additive to enhance redox status and improve meat quality of IUGR growing pigs via the Nrf2 signal pathway.

## 1. Introduction

Intrauterine growth retardation (IUGR) refers to growth lag, developmental restriction or impaired organs in fetuses. More than 15% of piglets and about 10% of newborn humans suffer from IUGR [1]. IUGR exhibits programming consequences, which is defined as permanent or long-term changes to the physiology, morphology or metabolism of a fetus, so that IUGR exerts permanent negative effects on postnatal growth and health [1,2,3]. Recent advances in research are focusing on the dynamics of IUGR fetal and offspring growth in order to predict the possible outcome and to optimize the time for treating or preventing IUGR [2,3,4]. Our previous study has documented that IUGR impairs antioxidant capacity and results in oxidative stress in weaning piglets [5]. Oxidative stress, including lipid and protein oxidation, negatively affects the entire quality traits of meat and decreases the nutritional values of meat products [6]. As for IUGR programming consequences, IUGR growing-finishing pigs might also show oxidative stress and affect the meat quality [2]. Dietary-supplemented antioxidants can improve antioxidant status, prevent lipid oxidation and improve pork quality [7]. However, an effective feeding strategy for preventing the impaired antioxidant capacity of IUGR pigs is still scanty. Considering the increasing demands for healthy food and worries about side effects of commonly used synthetic antioxidants (butylated hydroxyanisole, butylated hydroxytoluene and tertiary butyl hydroquinone), natural antioxidants from plants (herbs and spices) have started to receive more and more attentions recently [8,9]. 

Among natural antioxidants, turmeric, a rhizomatous herbaceous perennial plant (*Curcuma longa*), has received much interest in both culinary and medical fields [10]. Turmeric was widely used as a spice and traditional medicine in China and Southeast Asian countries. In the recent few decades, curcumin, the main natural polyphenol isolated from the rhizome of turmeric and other curcuma, exhibits a bright yellow color and possesses many great biological properties, such as antioxidant, anti-inflammatory and antimicrobial, and it has many beneficial effects on the management of the metabolic syndrome [11,12,13]. In broilers, we found that curcumin as a potential antioxidant could improve the oxidant stability of muscle and benefit meat quality [14]. In an IUGR rat model, we also found that dietary curcumin can attenuate inflammation, hepatic injury and oxidative damage [15]. Several studies suggest that the antioxidant mechanism of curcumin may be related to the nuclear factor erythroid 2-related factor 2 (Nrf2) signal pathway. For example, curcumin attenuates dimethylnitrosamine-induced liver injuries in rats via induction of the Nrf2-mediated upregulation of heme oxygenase-1 (HO-1) [16]. Curcumin has also been shown that it can alleviate heat-induced oxidative stress via the Nrf2/HO-1 pathway in Japanese quails [13]. Thus, we hypothesized that curcumin may improve the antioxidant capacity of IUGR growing pigs through the Nrf2 signal pathway, which in turn has the potential to improve meat quality. Therefore, in this study, we investigated the effects of dietary-added 200 mg/kg curcumin on IUGR pigs by determining the growth performance, meat quality and antioxidant capacity and further explored its potential mechanism via analysis of gene and protein expressions related to the Nrf2 signal pathway. These results may provide insights for the application of curcumin as a feed additive to raise IUGR pigs and improve meat quality. In addition, pigs are very similar to humans in terms of physiology and anatomy [1], which are commonly considered as ideal models for studying many diseases, so our results may also provide some useful theoretical references for IUGR offspring in humans.

## 2. Materials and Methods

### 2.1. Animal Care

All the procedures were carried out in accordance with the Chinese Guidelines for Animal Welfare and Experimental Protocol and were approved by the Institutional Animal Care and Use Committee of Nanjing Agricultural University, China (NJAU-CAST-2015-098).

### 2.2. Animals and Study Design

At the time of parturition (114 days gestation), 12 NBW (1.49 ± 0.02 kg) and 24 IUGR female (0.76 ± 0.02 kg) piglets (Duroc× Landrace× Large White) were selected from 12 sows with similar litter sizes. One NBW and two IUGR female piglets were chosen from each litter according to the previous studies [17,18,19]. All piglets were weaned at 26 d, and 24 IUGR piglets were randomly assigned into 2 groups (each group with 12 piglets). The 3 groups (12 NBW and 24 IUGR female piglets) were fed control diets supplemented 0 (NBW), 0 (IUGR) and 200 (IUGR + Cur) mg/kg curcumin from 26 to 115 days of age (n = 12). The effective content of curcumin (200 mg/kg) was selected according to the results of Lu et al. [20]. The composition and nutrient level of the basal diet was shown in the Appendix A.

During this feeding trial, pigs were housed on the concrete floors with water and feed ad libitum. After weaning, each group was housed in one pen, while 2 pigs were housed in one pen during 56–115 d. At the end of the feeding trial, 6 pigs (close to the average body weight) from each group were chosen. After fasting for 12 h, pigs were euthanized by electrical stunning and exsanguinated. The blood was centrifuged at 3000 *g* for 10 min at 4 °C and stored at −80 °C until analysis. The muscle (*biceps femoris*) of the left hind leg samples were frozen in liquid nitrogen and stored at −80 °C for further determination. For the meat quality, the muscle samples were stored at 4 °C. 

### 2.3. Growth Performance Analysis

The body weights were weighed at 0, 26, 56 and 115 d to calculate the average daily gain (ADG) during 0–26 d, 26–56 d and 56–115 d. The feed intake was measured during 56–115 d to calculate the average daily feed intake (ADFI) and feed:gain ratio (F:G) on a wet basis. 

### 2.4. Meat Quality Analysis

Muscle pH value was determined at 45 min (pH_45min_) and 24 h (pH_24h_) post-slaughter using a pH meter (PH-STAR, Mattuas, Germany), each sample with three replicates. The color of fresh meat was measured using a Minolta chromameter (CR-10, Konica Minolta Sensing, Inc., Osaka, Japan) with 8 mm measuring diameter and 8° illumination angle (CIE standard illuminant D65). The CIE L* (light index), a* (red index) and b* (yellow index) were collected from three different orientations on the cut surface of each leg muscle chop. Calibration was carried out prior to each color determination using a white standard plate. The drip loss was determined as described in our previous publication with some modifications [14]. Briefly, the leg muscle chops (size of 3 cm × 2 cm × 1 cm) were weighed, suspended and hooked in a sealed sample bag at 4 °C for 24 h and 48 h and then were weighed to calculate drip loss.

### 2.5. Assay of Antioxidant Capacity in Leg Muscle

The samples of biceps femoris were homogenized in ice-cold 0.90% sodium chloride buffer (w:v) as described by Zhang et al. [14]. The supernatant was obtained by centrifugation at 5000 *g* for 10 min at 4 °C. Activities of total antioxidant capacity (T-AOC), superoxide dismutase (SOD), catalase (CAT), peroxidase (POD), glutathione peroxidase (GPX) and the level of glutathione (GSH), thyrotropin (TSH), malondialdehyde (MDA) and protein carbonyls (PC) in the supernatant were determined with corresponding commercial kits purchased from the Nanjing Jiancheng Bioengineering Institute (Nanjing, China). The activities of T-AOC, SOD, CAT, POD and GPX were expressed as units (U) per milligram of protein. The concentrations of MDA and PC were expressed as nanomoles per gram of protein. The levels of TSH and GSH were expressed as millimoles and milligram per gram of protein, respectively.

### 2.6. Mineral Concentration in Leg Muscle

The mineral concentrations (Zn, Fe, Cu, Mn and Se) in the muscles were determined as the methods of Demirbaş et al. [21] and Wang et al. [22] with some modifications. Briefly, an acid mixture (HNO_3_:HClO_4_ = 4:1, v:v) was added to the muscle sample (about 1.0 g), stewing for 12 h at room temperature, and then was treated with high temperature until digested completely. The reaction condition of high temperature was as follows: 90 °C for 30 min; 120 °C for 30 min; 160 °C for 120 min and 210 °C for 210 min. The digest was diluted with 5% HNO_3_ solution to the optimal level for further determination. After the external matrix-matched standard curves were determined via corresponding standards, which were diluted with 5% HNO_3_ solution, the blanks and prepared solutions of samples were analyzed with inductively coupled plasma optical emission spectrometry (ICP-OES).

### 2.7. Determination of mRNA Expression

Total RNA from the leg muscle samples was extracted using Trizol Reagents (TaKaRa Biotechnology, Dalian, Liaoning, China). The integrity of RNA was examined by 1% agarose gel. The RNA concentration was determined, and its quality was verified (ratios of absorption including 260/280 nm and 263/230 nm between 1.90 and 2.05) via a NanoDrop ND-2000 UV spectrophotometer (NanoDrop Technologies, Wilmington, DE, USA). The RNA (each sample 2 μg) was reverse-transcribed into cDNA using PrimeScript™ RT Reagent Kit (TaKaRa Biotechnology, Dalian, Liaoning, China) according to the manufacturer’s protocol. The cDNA was stored at −20 °C for further determination. The specific primers for the target genes related to the antioxidant and Nrf2 signal pathway, including CAT, SOD, glutathione S-transferase (GST), HO-1, kelch-like ECH-associated protein 1 (Keap1), nuclear factor erythroid 2 related factor 2 (Nrf2), NAD (P)H dehydrogenase, quinone 1 (NQO1), glutamate-cysteine ligase catalytic subunit (GCLC), glutamate-cysteine ligase modifier subunit (GCLM) and gamma-glutamyltransferase 1 (GGT1), were synthesized by Invitrogen Biotech Co. Ltd. (Shanghai, China) and are listed in Table 1. GAPDH was used as a house-keeping gene to normalize expressions of the target genes. Quantitative real-time polymerase chain reaction (qRT-PCR) was performed with ABI StepOnePlus real-time PCR system (Applied Biosystems, Life technologies Corporation^TM^, Grand Island, NY, USA) according to the manufacturer’s guidelines. The SYBR Green PCR reaction system was 10 μL in total, which was composed of 5 μL ChamQ SYBR qPCR Master Mix (2×), 0.2 μL forward and reverse primers, 0.2 μL ROX Reference Dye 1 (50×), 1 μL cDNA and 3.6 μL ddH_2_O. The relative mRNA expressions of target genes were calculated using the 2^−ΔΔCt^ method as previously reported by Livak et al. [23] and normalized to the NBW group.

### 2.8. Western Blotting Anlysis

The Western blotting analysis was conducted according to our previous method with some modifications [24]. Briefly, the leg muscle samples were weighed and homogenized in 1:10 (w/v) in 10 mM Tris–HCl buffer (pH = 7.4), and protein levels were determined with the BCA kit, which was provided by Beyotime (Jiangsu, China). Subsequently, about 30 μg protein from each sample was separated via sodium dodecyl sulfate-polyacrylamide gel electrophoresis and transferred to polyvinylidene difluoride membranes. The membranes were washed twice for 10 min in Tris-buffered saline Tween (TBST) solution, blocked with blocking buffer (5% nonfat dry milk) for 1.5 h at room temperature and incubated with the antibodies against Nrf2 (1:600 dilution; Proteintech Group, Inc., Rosemont, IL, USA); Keap1 (1:600 dilution; Proteintech Group, Inc., Rosemont, IL, USA) and NQO1 (1:4 000 dilution; Proteintech Group, Inc., Rosemont, IL, USA) and α-Tubulin (1:1000 dilution; Proteintech Group, Inc., Rosemont, IL, USA) for 16 h at 4 °C. After washing three times with TBST solution, the membranes were incubated with the secondary antibody (horseradish-peroxidase-conjugated goat anti-rabbit Ig G; 1:4000 dilution; Proteintech Group, Inc., Rosemont, IL, USA). The blots were detected using enhanced chemiluminescence reagents (Beyotime, Jiangsu, China) followed by autoradiography. Photographs were taken using the Luminescent image analyzer LAS-4000 system (Fujifilm, Tokyo, Japan), and the antigen-antibody complexes were quantified by the Quantity One software (Bio-Rad Laboratories, Hercules, CA, USA).

### 2.9. Statistical Analysis

To estimate the effects of the treatments on the response variables evaluated, a one-way ANOVA was used. To evaluate the difference among groups, multiple comparisons were conducted using the Duncan test. Pens were used as experimental units for the analysis of ADFI and F:G, and for other parameters, each pig was used as the experimental unit. Statistical significance was considered at *p* < 0.05. All statistic procedures were done in the SPSS (20.0) package.

## 3. Results

### 3.1. Growth Performance

As indicated in Table 2, compared with the NBW group, IUGR pigs had significantly decreased body weight on 0 d, 26 d and 56 d (*p* < 0.01), but no significant difference on 115 d was found among the NBW, IUGR and IUGR + Cur groups (*p* > 0.05). Similarly, IUGR pigs had lower ADG (*p* < 0.05) during 0–26 d (*p* < 0.01) and 26-56 d (*p* = 0.02) than NBW pigs, while they showed no significant difference during 56-115 d (*p* > 0.05). Dietary-added 200 mg/kg curcumin did not affect the body weights, ADG and F:G of IUGR pigs (*p* > 0.05); however, there was significantly decreased ADFI during 56–115 d than both the NBW and IUGR groups (*p* < 0.01). 

### 3.2. Meat Quality

Effects of dietary curcumin on meat quality in leg muscles are presented in Table 3. Results showed that IUGR did not significantly affect the muscle pH, drip loss and meat color (L*, a* and b*) of pigs as compared to the NBW group (*p* > 0.05). However, compared with the NBW and IUGR groups, a significant decrease was observed in drip loss at 24 h and 48 h in the IUGR + Cur group (*p* < 0.01). In addition, the a* value of pigs in the IUGR + Cur group was significantly increased as compared to the NBW group (*p* = 0.02).

### 3.3. Antioxidant Enzyme Activity, MDA and Protein Carbonyls (PC)

Effects of dietary curcumin on antioxidant capacity of leg muscles in IUGR growing pigs are shown in Table 4. Results showed that IUGR pigs had higher MDA (*p* < 0.01) and PC (*p* = 0.03) levels in leg muscles and had significantly lower GPX (*p* = 0.02), CAT (*p* < 0.01) and POD (*p* = 0.02) activities than NBW pigs but did not show significant difference in GSH and TSH contents (*p* > 0.05). Compared with the IUGR group, dietary-added 200 mg/kg curcumin significantly decreased MDA (*p* < 0.01) and PC (*p* = 0.03) contents in the muscles and increased the CAT (*p* < 0.01), SOD (*p* = 0.04) and POD (*p* = 0.02) activities. There were no significant differences in antioxidant enzyme activities (GPX, CAT, T-AOC, SOD and POD) and contents of MDA, PC, GSH and TSH between the NBW and IUGR + Cur groups (*p* > 0.05).

### 3.4. Mineral Concentration in Leg Muscles

Effects of curcumin on mineral concentrations of leg muscles in IUGR growing pigs are summarized in Table 5. Results showed that there were no significant differences in mineral concentrations of Zn, Fe, Cu, Mn and Se among the NBW, IUGR and IUGR+Cur groups (*p* > 0.05).

### 3.5. Gene Expression

As shown in Figure 1, IUGR pigs had lower mRNA expressions of HO-1 and CAT as compared with NBW pigs (*p* < 0.05). Compared with the IUGR group, dietary-supplemented 200 mg/kg curcumin significantly increased mRNA expressions of GST, HO-1 and CAT of leg muscles (*p* < 0.05), while it did not show significant effects on SOD mRNA expression (*p* > 0.05). There were no significant differences in mRNA expressions of SOD, GST, HO-1 and CAT in leg muscles of pigs between the IUGR + Cur and NBW groups (*p* > 0.05).

### 3.6. Protein Expressions of Keap1/Nrf2 Signal Pathway in Leg Muscle

Effects of curcumin on protein expressions of the Keap1/Nrf2 signal pathway in leg muscles are shown in Figure 2. Results showed that there were no significant differences in protein expressions of Keap1, Nrf2 and NQO1 between the NBW and IUGR groups (*p* > 0.05). However, pigs in the IUGR + Cur group had significantly higher Nrf2 protein expression than those of the NBW group (*p* < 0.05) and significantly higher NQO1 protein expression than those of the IUGR group (*p* < 0.05).

### 3.7. Gene Expressions of the Keap1/Nrf2 Signal Pathway in Leg Muscles

Effects of curcumin on gene expressions of the Keap1/Nrf2 signal pathway in leg muscles, including Keap1, Nrf2, NQO1, GCLC, GCLM and GGT1, are shown in Figure 3. Results showed that there were no significant differences in the mRNA expressions of Keap1, Nrf2, NQO1, GCLC, GCLM and GGT1 between the NBW and IUGR groups (*p* > 0.05). However, pigs in the IUGR + Cur group had significantly higher mRNA expressions of Nrf2, NQO1 and GGT1 than those of the IUGR group (*p* < 0.05) and significantly increased GCLM and GGT1 mRNA expressions than those of the NBW group (*p* < 0.05). There were no significant differences in the Keap1 and GCLC mRNA expressions among the three groups (*p* > 0.05).

## 4. Discussion

IUGR can result in low birth weights and show postnatal catch-up growths after birth, which are considered to be important risk factors for later developments of chronic metabolic diseases [25,26]. These findings are in agreement with the results of our present study, in which IUGR pigs exhibited significantly decreased ADG during 0–56 d and lower body weights on 0 d, 26 d and 56 d but had no significant difference in body weights on 115 d and had no difference in ADG during 56–115 d between the IUGR and NBW groups. In consistent with our present study, Attig et al. reported that IUGR pigs showed reduced body weights until 60 d while exhibiting catch-up growth and had no significant difference with NBW pigs after 105 d [27]. Poore found that IUGR pigs showed postnatal catch-up developments and had similar body weights with NBW finishing pigs [28]. Similar effects were also reported in IUGR rat [29], mice [30] or lamb [31] models. In this study, results also showed that diets supplemented with 200 mg/kg curcumin significantly decreased the ADFI, while they did not significantly affect the body weights, ADG and F:G ratio of the IUGR pigs. The possible reason might be that curcumin can enhance digestive tract functions and can increase the relative weight gain with decreased ADFI. For example, Xun et al. reported that diets added with 300 mg/kg or 400 mg/kg curcumin improve the intestinal mucosal barrier integrity, morphology and immune status of weaned pigs [32]. Jiang et al. also find that diets supplemented with 5 g/kg curcumin significantly increased trypsin and lipase activities in the intestines of fish [33]. Most IUGR catch up in growth and attain normal body weight, but still show permanent negative effects on health, such insulin resistance and type 2 diabetes [2,3]. The consequences of IUGR programming and related mechanisms should be further studied. 

IUGR may also affect meat quality as well as growth performance via causing changes in the physiology and metabolism [34,35]. In our present study, the fresh meat quality, including pH values (pH_45min_ and pH_24h_); drip loss (24 and 48 h) and meat color (L*, a* and b* values), were determined. Results indicated that IUGR showed no significant effects on these parameters of meat quality; however, pigs in the IUGR+Cur group had higher a* values as compared to that in the NBW group, which was in agreement with the results of Zhang et al., who reported that dietary-supplemented 0–200 mg/kg curcumin showed a significant linear and quadratic increase in a* values of breast muscles in broilers [14]. Mancini et al. verified that supplemented 3.5 g of turmer power/100 g meat could increase the a* values and delay the rate of decreasing in a* values of burgers with aging [36]. In our present study, we also found that dietary-supplemented 200 mg/kg curcumin significantly decreased the drip loss at both 24 h and 48 h. In agreement with our results, Zhang et al. found that diets supplemented with 100 mg/kg curcumin for 21 d significantly decreased the drip loss of breast muscles in broilers at 48 h [14]. 

The decreased drip loss might be related to enhanced antioxidant capability and reduced oxidative stress [6,14]. Oxidative stress is one of the major causes of meat quality deterioration, including the development of autoxidation off-flavor, formation of toxic compounds, poor shelf life and nutrient and drip losses [6]. Oxidative stress is mainly accountable for the dysfunction of redox regulation involving reactive oxygen species (ROS) and reactive nitrogen species (RNS), which could lead to the oxidation of lipid and proteins and damage of DNA and cells [37,38]. MDA and PC contents are main markers of endogenous lipid peroxidation and protein oxidation, respectively [39,40]. In the current study, increased MDA and PC concentrations in leg muscles of the IUGR pigs indicated that IUGR may lead to both lipid and protein oxidation, while dietary-added 200 mg/kg curcumin could protect pigs from IUGR-induced oxidation, since the MDA and PC concentrations were decreased to normal levels of the NBW pigs. Normally, the antioxidant system, including nonenzymatic and enzymatic antioxidant capacity, can scavenge excessive free radicals and keep a balance between oxidation and antioxidation [9]. Therefore, to evaluate the effects of the IUGR and dietary curcumin on the antioxidant defense system of pigs, we determined concentrations of GSH and TSH and T-AOC, GPX, CAT, SOD and POD activities in leg muscles. Results showed that the IUGR pigs had lower enzymatic antioxidant (GPX, CAT, and POD) activities than the NBW pigs, and dietary-added 200 mg/kg curcumin significantly increased the enzymatic antioxidant (CAT, SOD and POD) activities in leg muscles but did not show significant effects on the nonenzymatic capacity (contents of GSH and TSH) and T-AOC. Enzymes of SOD, CAT, GPX and peroxidases serve as front-line antioxidant defenses, which can catalyze the dismutation of O^2−^ to H_2_O_2_ and further reduce to H_2_O, while GSH and TSH concentrations and T-AOC reflect the nonenzymatic antioxidant defense system [41]. Therefore, our present study suggested that IUGR induced oxidative stress, and the beneficial effects of curcumin on the antioxidant defense system in leg muscles of growing pigs are mainly related to the enzymatic antioxidant defense system instead of the nonenzymatic system. 

Mineral concentrations in serum and tissues, including Zn, Fe, Cu, Mn and Se, are closely involved in the enzymatic antioxidant defense system, since they are important components of many antioxidant enzymes or act as enzyme activators [37]. For example, Fe plays important roles in the redox status as components of peroxidases and CAT, while Cu, Mn and Zn are components of SOD. The beneficial effects of dietary Se on the antioxidant system can partially equal to vitamin E in both pigs and broilers. A previous study also documented that IUGR could impair antioxidant capacity and lead to oxidative damage in weaned piglets, which may be associated with the decreased concentrations of these redox-active trace minerals in tissues [5]. Zadrożna et al. showed the IUGR placentas had reduced concentrations by 23% and Zn by 37%, which showed a strong relation with Cu-Zn SOD activities [42]. However, in our current study, the concentrations of Zn, Fe, Cu, Mn and Se in leg muscles were not affected by both IUGR and dietary-added 200 mg/kg curcumin, which might contribute to the catch-up growth in the IUGR growing pigs. However, more research is still needed.

To further explore effects of IUGR-induced oxidative stress and the beneficial effects of dietary curcumin, the mRNA expressions of SOD, GST, HO-1 and CAT in leg muscles were analyzed. Results showed that the IUGR pigs had lower mRNA expressions of HO-1 and CAT, while dietary-supplemented 200mg/kg curcumin significantly increased GST, HO-1 and CAT mRNA expressions. In consistent with our present study, Ferencz et al. reported that the mRNA expressions of HO-1, HO-2 and CAT in low birth weight neonates were considerably lower than that in high birth weight neonates [43]. Our previous studies in rat or mice models also found that dietary-added curcumin can increase the mRNA expressions of GST and CAT to alleviate oxidative stress and inflammation [15,44].

Additionally, mRNA expressions of some antioxidant genes such as HO-1 are closely related to the Keap1/Nrf2 pathway [16,45]. Nrf2 is an essential transcription factor that regulates an array of antioxidant defense genes [46]. Keap1 is a negative regulator of Nrf2 [47]. Upon stimulation by inducers, Nrf2 dissociates from Keap1 and translocates into the nucleus and binds to antioxidant response elements (ARE) and activates a battery of highly specialized gene and protein expressions, including NQO1, HO-1, GCLC, GCLM and GGT1 [16,48,49]. Therefore, expressions of proteins (Keap1, Nrf2 and NQO1), and genes (Keap1, Nrf2, NQO1, GCLC, GCLM and GGT1) related to the Keap1/Nrf2 pathway were determined in our present study to further investigate the molecular mechanism of IUGR-induced oxidative stress and the beneficial effects of dietary curcumin. Results showed that there were no significant differences in expressions of these proteins and genes between the IUGR and NBW groups, which suggested that IUGR-induced oxidative stress in leg muscles of growing pigs might not be related to the Keap1/Nrf2 signal pathway. However, dietary-supplemented 200 mg/kg curcumin significantly enhanced NQO1 protein expression and increased the mRNA expressions of Nrf2, NQO1 and GGT1 as compared with the IUGR group. These results suggested that beneficial effects of dietary curcumin on IUGR-induced oxidative stress may be, at least in part, attributed to the enhanced Nrf2 signal pathway, which was consistent with Farombi and Shrotriya [16]. Sahina et al. has also documented that curcumin ameliorates heat stress in quail via the inhibition of oxidative stress, in which Nrf2 and HO-1 expressions are enhanced as supplemented curcumin levels increase [13]. Similarly, Receno et al. in an aging rat model found that dietary curcumin supplementation augments Nrf2 nuclear translocation and attenuates oxidative stress in the skeletal muscle [50]. 

## 5. Conclusions

Our results indicated that IUGR pigs exhibited growth inhibitions before 56 d and showed postnatal catch-up growth during 56–115 d. In addition, IUGR could induce oxidative stress in leg muscles, and dietary-supplemented 200 mg/kg curcumin could improve meat color, increase water-holding capacity and attenuate the oxidative stress of pigs via the Nrf2 signal pathway. This study suggested that curcumin could serve as a potential natural antioxidant in nutrition interventions of IUGR offspring to improve the redox status of leg muscles and benefit the meat quality.

## Figures and Tables

**Figure 1 animals-10-00539-f001:**
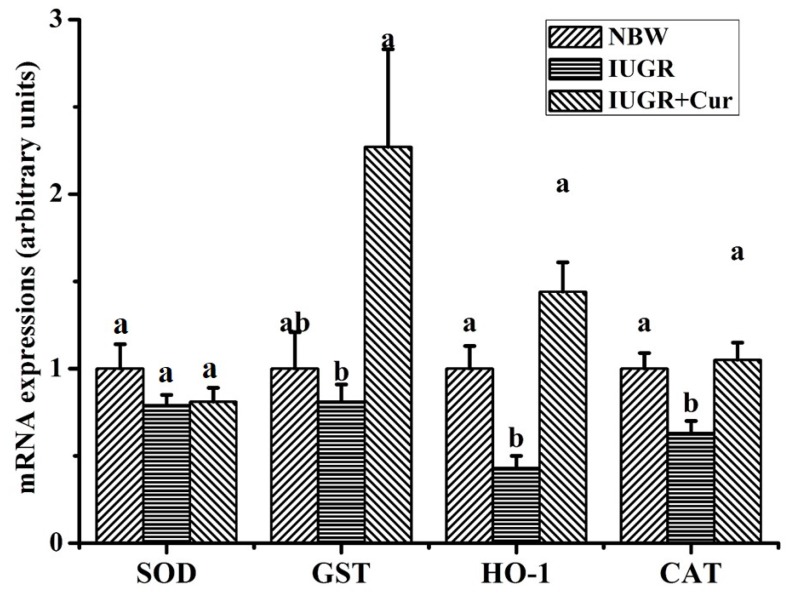
Effects of curcumin on antioxidant-related gene expressions in leg muscles. Data were normalized to the normal birth weight (NBW) group and expressed as means with their standard errors (n = 6). ^a,b^ Means that values of the same parameter with different superscripts were significantly different (*p* < 0.05). SOD, superoxide dismutase; GST, glutathione S-transferase; HO-1, heme oxygenase-1; CAT, catalase; IUGR, intrauterine growth retardation and IUGR + Cur, IUGR pigs given a control diet supplemented with 200 mg/kg curcumin.

**Figure 2 animals-10-00539-f002:**
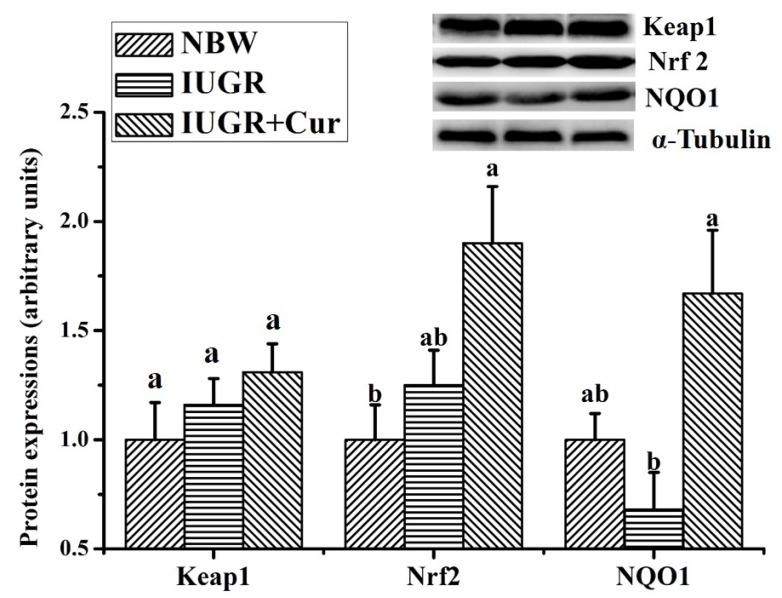
Effects of curcumin on protein expressions of the kelch-like ECH-associated protein 1 (Keap1)/Nrf2 signal pathway in leg muscles. Data were expressed relative to α-Tubulin and normalized to the NBW group and expressed as means with their standard errors (n = 4). ^a,b^ Means that values of the same parameter with different superscripts were significantly different (*p* < 0.05). NQO1, NAD (P)H dehydrogenase, quinone 1.

**Figure 3 animals-10-00539-f003:**
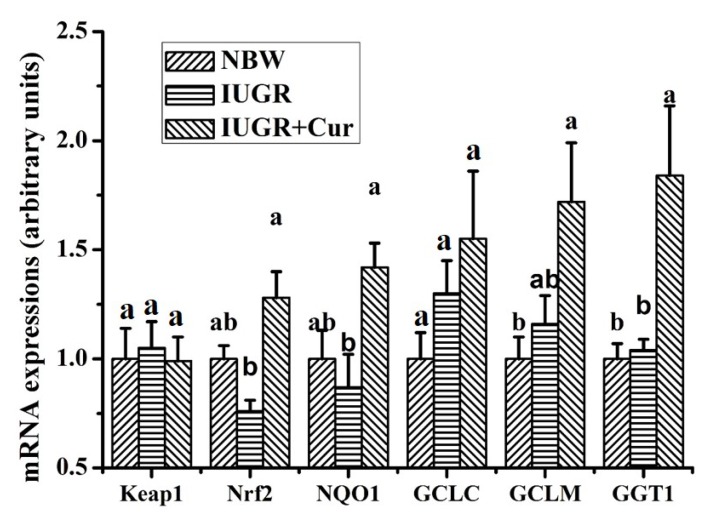
Effects of curcumin on gene expressions of the Keap1/Nrf2 signal pathway in leg muscles. Data were normalized to the NBW group and expressed as means with their standard errors (n = 6). ^a,b^ Means that values of the same parameter with different superscripts were significantly different (*p* < 0.05). GCLC, glutamate-cysteine ligase catalytic subunit; GCLM, glutamate-cysteine ligase modifier subunit and GGT1, gamma-glutamyltransferase 1.

**Table 1 animals-10-00539-t001:** Primer sequences used in quantitative real-time PCR assays. Keap1, kelch-like ECH-associated protein 1; CAT, catalase; GST, glutathione S-transferase; SOD, superoxide dismutase; HO-1, heme oxygenase-1; NQO1, NAD (P)H dehydrogenase, quinone 1; GCLC, glutamate-cysteine ligase catalytic subunit; GCLM, glutamate-cysteine ligase modifier subunit and CGT1, gamma-glutamyltransferase 1.

Gene	Accession No.	Sequence (5′-3′)	Size (bp)
GAPDH	NM_001206359.1	F-CGTCCCTGAGACACGATGGT	194
		R-GCCTTGACTGTGCCGTGGAAT	
Keap1	XM_021076667.1	F: CGTGGAGACAGAAACGTGGA	239
		R: CAATCTGCTTCCGACAGGGT	
Nrf2	NM_001114671.1	F: GACAAACCGCCTCAACTCAG	183
		R: GTCTCCACGTCGTAGCGTTC	
CAT	XM_021081498.1	F:AGCTTTGCCCTTGCACAAAC	119
		R:ACATCCTGAACAAGAAGGGGC	
GST	NM_214389.2	F:TTCAATGGCCGAGGCAGAAT	216
		R: GAGGTTGTACTTGGTGGCGA	
SOD	NM_001190422.1	F: CATTCCATCATTGGCCGCAC	118
		R: TTACACCACAGGCCAAACGA	
HO-1	NM_001004027.1	F: CAAGCAGAAAATCCTCGAAG	241
		R: GCTGAGTGTCAGGACCCATC	
NQO1	NM_001159613.1	F: GATCATACTGGCCCACTCCG	200
		R: GAGCAGTCTCGGCAGGATAC	
GCLC	XM_003482164.4	F:GGCGACGAGGTGGAATACAT	123
		R: GTTTGGGTTTGTCCTTTCCCC	
GCLM	XM_001926378.4	F: GCATCTACAGCCTTACTGGGA	180
		R: GTTAAATCGGGCGGCATCAC	
GGT1	NM_214030.1	F:ATCACACCAGGAAAACAGCCA	118
		R: CGGTAGATGTGGTGATCTGTGT	

**Table 2 animals-10-00539-t002:** Effects of curcumin on the growth performance of IUGR growing pigs.

Item ^1^	NBW	IUGR	IUGR + Cur	*p*
Body weight(Kg)				
0 d	1.49 ± 0.02 ^a^	0.75 ± 0.01 ^b^	0.76 ± 0.01 ^b^	<0.01
26 d	7.52 ± 0.35 ^a^	5.76 ± 0.20 ^b^	6.05 ± 0.10 ^b^	<0.01
56 d	17.03 ± 0.42 ^a^	13.39 ± 0.46 ^b^	13.96 ± 0.29 ^b^	<0.01
115 d	57.53 ± 1.31	53.46 ± 1.80	53.52 ± 1.53	0.12
ADG(Kg)				
0-26 d	0.25 ± 0.01 ^a^	0.20 ± 0.01 ^b^	0.20 ± 0.01 ^b^	<0.01
26-56 d	0.30 ± 0.01 ^a^	0.25 ± 0.02 ^b^	0.26 ± 0.01 ^b^	0.02
56-115 d	0.69 ± 0.02	0.68 ± 0.03	0.67 ± 0.03	0.91
ADFI(Kg/d)				
56-115 d	2.35 ± 0.02 ^a^	2.31 ± 0.02 ^a^	2.17 ± 0.05 ^b^	<0.01
F:G				
56-115 d	3.43 ± 0.08	3.41 ± 0.08	3.25 ± 0.12	0.40

^1^ NBW, normal birth weight group given a control diet; IUGR, intrauterine growth retardation group given a control diet; IUGR + Cur, IUGR pigs given a control diet supplemented with 200 mg/kg curcumin; ADG, average daily gain; ADFI, average daily feed intake and F:G, feed-to-gain ratio. Data were expressed as mean ± SE for BW and ADG, n = 12; for ADFI and F:G, n = 6. ^a, b^ Means that values within a row with different superscript letters were significantly different (*p* < 0.05).

**Table 3 animals-10-00539-t003:** Effects of curcumin on the meat quality of IUGR growing pigs.

Item	NBW	IUGR	IUGR+Cur	*p*
pH				
pH_45min_	6.28 ± 0.07	6.39 ± 0.08	6.33 ± 0.11	0.70
pH_24h_	6.26 ± 0.01	6.21 ± 0.04	6.25 ± 0.03	0.42
Drip loss				
24 h	0.18 ± 0.01^a^	0.20 ± 0.01^a^	0.14 ± 0.01^b^	<0.01
48 h	0.22 ±0.01^a^	0.25 ± 0.01^a^	0.17 ± 0.01^b^	<0.01
Meat color				
L*	45.10 ± 1.58	44.90 ± 0.78	41.66 ± 0.79	0.08
a*	9.75 ± 0.30^b^	9.90 ± 0.28^ab^	11.25 ± 0.49^a^	0.02
b*	17.50 ± 0.62	17.70 ± 0.67	17.35 ± 0.77	0.94

Data were expressed as mean ± SE (n = 6). ^a, b^ Means that values within a row with different superscript letters were significantly different (*p* < 0.05).

**Table 4 animals-10-00539-t004:** Effects of curcumin on antioxidant capacity of leg muscles in IUGR growing pigs.

Item ^1^	NBW	IUGR	IUGR + Cur	*p*
MDA(nmol/mgprot)	3.08 ± 0.26 ^b^	4.65 ± 0.32 ^a^	3.42 ± 0.19 ^b^	<0.01
PC (nmol/mgprot)	3.19 ± 0.16 ^b^	3.85 ± 0.17 ^a^	3.43 ± 0.15 ^ab^	0.03
GSH(mg/gprot)	0.87 ± 0.07	0.94 ± 0.04	1.03 ± 0.11	0.41
GPX (U/mgprot)	16.65 ± 1.67 ^a^	13.65 ± 0.53 ^b^	15.50 ± 0.82 ^ab^	0.02
CAT(U/mgprot)	18.12 ± 1.06 ^a^	14.18 ± 0.32 ^b^	18.21 ± 0.97 ^a^	<0.01
T-AOC(U/mgprot)	0.28 ± 0.01	0.28 ± 0.01	0.29 ± 0.01	0.94
SOD(U/mgprot)	17.38 ± 0.34 ^ab^	16.69 ± 0.23 ^b^	17.83 ± 0.28 ^a^	0.04
TSH(mmol/gprot)	0.13 ± 0.01	0.12 ± 0.01	0.12 ± 0.01	0.41
POD(U/mgprot)	0.27 ± 0.02 ^a^	0.19 ± 0.02 ^b^	0.29 ± 0.03 ^a^	0.02

^1^ MDA, malondialdehyde; PC, protein carbonyls; GPX, glutathione peroxidase; CAT, catalase; T-AOC, total antioxidant capacity; SOD, superoxide dismutase; POD, peroxidase; GSH, glutathione and TSH, thyrotrophin. ^a,b^ Means that values within a row with different superscript letters were significantly different (*p* < 0.05).

**Table 5 animals-10-00539-t005:** Effects of curcumin on mineral concentrations of leg muscles in IUGR growing pigs.

Item	NBW	IUGR	IUGR + Cur	*p*
Zn (mg/kg)	15.71 ± 1.84	11.50 ± 0.91	13.42 ± 1.85	0.21
Fe (mg/kg)	100.47 ± 10.11	92.50 ± 9.65	89.70 ± 9.26	0.72
Cu (mg/kg)	1.04 ± 0.09	1.10 ± 0.09	1.13 ± 0.05	0.72
Mn (mg/kg)	0.70 ± 0.06	0.73 ± 0.02	0.75 ± 0.06	0.80
Se (mg/kg)	0.57 ± 0.04	0.59 ± 0.04	0.58 ± 0.02	0.87

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
