# Peer review of "Dietary Supplemented Curcumin Improves Meat Quality and Antioxidant Status of Intrauterine Growth Retardation Growing Pigs via Nrf2 Signal Pathway"

_animals, 2020, doi:10.3390/ani10030539_

Round 1

Reviewer 1 Report

In the study, authors tested the hypothesis that dietary supplementation of curcumin may protect growing pigs from IUGR induced oxidative stress. It is a very interesting topic in terms of IUGR, oxidative stress, natural antioxidant. However, there are some critical concerns as followed.

  1. Data on drip loss, pH value, and fresh meat color a* and b* did not stay in the proper numerical range. I cannot imagine what wrong happened in the measurement.
  2. There no obvious changes in nrf2 protein expression among treatments. Even as authors claimed, nrf2 expression was increased in IUGR group compared with that of NBW groups, and then what meaning of increased nrf2 protein expression in IUGR + curcumin relative to NBW is it?  The study should include changes in phosphorylation of nrf2 among treatments to draw the conclusion.

Minor errors:

  1. In introduction part, reference 7 that holds on the opposite opinion was used improperly.
  2. In materials and methods, the piglets were not randomly assigned into 3 groups.
  3. The indices concerning meat quality were measured in growing pigs rather than finishing pigs. In fact, it does not make sense. Authors should explain it reasonably
  4. The SE of T-AOC(U/mgprot) is only 0.01, it seemed curious, please check.
  5. English writing should be improved, for instance, “benefit meat quality” is not typical expression way in English.

Reviewer 2 Report

Major Comments

The present study negotiates an interesting topic for the pig industry due to the hyperprolific sows and the increased number of piglets at birth. Overcoming problems of low-birth weight piglets and postnatal growth is under debate. It should be further elaborated by the authors, the fact that despite final slaughter weight was similar, still antioxidant status was different. The compensation of body-weight growth, could be associated with a recovery also in antioxidant mechanisms. In infants for example samples were taken before parturition, close to the time point in which body weight difference are more significant. In the current study, approximately 50 days elapse between the time point of significant differences in bodyweight between treatments and sampling at slaughter. Furthermore, the composition of diets fed to the pigs during the different growth phases is missing. This is vital information, especially regarding any pro-oxidant material used e.g. specific fat source. Also the authors do not mention if the oxidative status of the diets was measured.

Minor comments

Line numbering is not provided, which makes difficult to address specific comments

In title IUGR should be defined.

Correct hurbs in introduction

Animals and Study Design: please refer to previous comment regarding diet composition and analysis

Growth Performance Analysis: how the pigs from each treatment were housed? Which was considered the experimental unit? The pen or the animal? Also feed intake was calculated on a pen basis?

Statistical Analysis: similar comment as previous. The experimental unit for each parameter should be defined.

In figures maybe the same superscript could be placed in columns with no significant difference between the treatments

“The possible reason might be that curcumin can enhance digestive tract function and could increase the relative weight gain with decreased ADFI” please elaborate on the potential mechanism, e.g. improved secretion of digestive enzymes? Also a potential negative palatability effect of curcumin could be involved?

Reviewer 3 Report

Key words are repeated in the title.

Please give information about composition of the feed mixtures.

Reviewer 4 Report

The manuscript has minor corrections. See adjunted file.

Reviewer 5 Report

This experiment studied effects of dietary curcumin on the growth performance, meat quality and antioxidant status of IUGR growing pigs, and further explored the Nrf2 signal pathway via RT-PCR and western blotting technology. Those results showed that dietary supplemented 200 mg/kg curcumin could improve meat quality and alter the expression of some genes and proteins that related to Nrf2 signal pathway. This experiment and results are interesting and this manuscript is also well organized. Generally speaking, the content of this manuscript is recommended for publication. However, there are some parts which should be corrected carefully and it is better for the authors to find a native English speaker to further revise this manuscript. The mainly parts needed to be corrected are listed as follow:

  1. This manuscript should be read carefully and more attentions should be paid to some minor errors. For example:
  • Line 21, change “intrauterine growth retardaation (IUGR)” into “IUGR”;
  • Line 25, change “humans” into “human”
  • Line 42. Change “P<0.05” to “P<0.05” and please correct the others in this manuscript.
  • Line 107, it’s better to change “the feed trial” into “this feeding trial”;
  • Line 175, change “the leg muscle sample” into ““the leg muscle samples”
  • Line 237, change “IUGR pigs given the control diets supplemented with 200 mg/kg;” into “IUGR pigs given the control diets supplemented with 200 mg/kg curcumin;”
  1. Line 143, it’s better to add more details about “treated with high temperature”;
  2. Line 286, it’s better for the author to read the sentence (a,bmeans that the same parameter with different superscripts are significantly different (P<0.05).) carefully and try to revise it.
  3. Pay attention to the format of references, such as Line 105-106 (Reference 20), this refernce should be check it carefully; please check the reference 5, 9, 10, 49 and 50, and correct the format follow the instrutions of the journal.

Round 2

Reviewer 1 Report

I have no further comment. 

Reviewer 2 Report

The authors have complied with the majority of the suggested corrections/additions in the revised version.

Minor spelling mistakes should be checked before final acceptance of the manuscript.